# Sleep Traits Causally Affect the Brain Cortical Structure: A Mendelian Randomization Study

**DOI:** 10.3390/biomedicines11082296

**Published:** 2023-08-18

**Authors:** Yanjing Chen, Shiyi Lyu, Wang Xiao, Sijie Yi, Ping Liu, Jun Liu

**Affiliations:** 1Department of Radiology, Second Xiangya Hospital, Central South University, Changsha 410011, China; 218211070@csu.edu.cn (Y.C.); lvshiyi1112@163.com (S.L.); yisijie@csu.edu.cn (S.Y.); 218212365@csu.edu.cn (P.L.); 2Department of General Surgery, Second Xiangya Hospital, Central South University, Changsha 410011, China; xiaowang1401751721@163.com; 3Clinical Research Center for Medical Imaging in Hunan Province, Changsha 410011, China

**Keywords:** brain cortical surface area (SA), cortical thickness (TH), sleep, brain cortical structure, mendelian randomization

## Abstract

**Background**: Brain imaging results in sleep deprived patients showed structural changes in the cerebral cortex; however, the reasons for this phenomenon need to be further explored. **Methods**: This MR study evaluated causal associations between morningness, ease of getting up, insomnia, long sleep, short sleep, and the cortex structure. **Results**: At the functional level, morningness increased the surface area (SA) of cuneus with global weighted (beta(b) (95% CI): 32.63 (10.35, 54.90), *p* = 0.004). Short sleep increased SA of the lateral occipital with global weighted (b (95% CI): 394.37(107.89, 680.85), *p* = 0.007. Short sleep reduced cortical thickness (TH) of paracentral with global weighted (OR (95% CI): −0.11 (−0.19, −0.03), *p* = 0.006). Short sleep reduced TH of parahippocampal with global weighted (b (95% CI): −0.25 (−0.42, −0.07), *p* = 0.006). No pleiotropy was detected. However, none of the Bonferroni-corrected *p* values of the causal relationship between cortical structure and the five types of sleep traits met the threshold. **Conclusions**: Our results potentially show evidence of a higher risk association between neuropsychiatric disorders and not only paracentral and parahippocampal brain areas atrophy, but also an increase in the middle temporal zone. Our findings shed light on the associations of cortical structure with the occurrence of five types of sleep traits.

## 1. Introduction

Numerous epidemiological studies have shown an association between sleep deprivation (insufficient sleep duration or quality) and poorer health outcomes, and the Centers for Disease Control and Prevention has identified sleep deprivation as a public health epidemic. (www.cdc.gov/features/dssleep/, accessed on 18 June 2023). Too little (<6 h) or too much (>9 h) sleep was also associated with significant cognitive deficits as well as more severe depressive symptoms, and higher body mass index (BMI) [1]. Persistent sleep disorders also increase the likelihood of cognitive decline [2] and Alzheimer’s Disease (AD) [3]. The intervention research also emphasized that sleep characteristics should be considered as potentially modifiable determinants of physical and mental health [4]. Although sleep is still considered a lifestyle trait, emerging evidence suggests that sleep duration and quality are to some extent genetic traits [5]. Therefore, it is important to identify independent risk factors for sleep characteristics and the patients who should be treated.

Cortical thickness (TH) is considered neuroimaging biomarkers for predicting cognitive decline and measurements of cortical surface area (SA) and its TH are a good reflection of cortical characteristics and have a high heritability [6]. Observational studies have reported [7] that alcohol use disorders make structural changes in the gray matter of the brain that lead to sleep disorders, but the directionality between the gray matter of the brain and sleep disorders is unknown. Moreover, previous studies have found that sleep deprivation may lead to a reduction in the cerebral cortex [8]. In addition, in a community sample, self-reported sleepiness on the Epworth Sleepiness Scale (ESS) score was associated with larger cortical gray matter volume and less prevalence of ischemic occult cerebral infarction as well as better cognitive performance, suggesting a healthier brain. However, this was only in men and non-carriers of apolipoprotein E ε4 (APOE ε4) [9]. Thus, this also suggests that alterations in sleep characteristics may also lead to alterations in the cerebral cortex. Sleep may serve as a biomarker of structural changes in the cerebral cortex [7] for prediction of the prognostic assessment of sleep-related disease treatment, such as AD [10], psychiatric disorders [11], alcohol use disorders [7], and chronic kidney disease [12]. However, the underlying genetic and environmental factors associated with sleep and structural changes in the brain are still poorly understood, and whether changes in sleep habits precede reductions in GM require further study. 

The gold standard for exploring cause-and-effect relationships is the randomized controlled trial. Theoretically, the effect of sleep habits on the development of cortical complications could be assessed by randomizing the subjects participating in a study into equal groups through interventions that maintain different sleep states. However, such trials lack more reliable and trustworthy data due to small sample sizes, ethical reasons, and the complex relationship between sleep habits and the cerebral cortex.

GWAS (Genome-Wide Association Study) is a method that investigates the associations between the human genome and specific traits, such as disease risk. It involves analyzing genomic data from a large number of individuals to identify the relationship between DNA variations and specific traits. Follow up analysis of GWAS allows us to gain a deeper understanding of the genetic relationships between traits.

Mendelian randomization (MR) treats genetic variants as instrumental variables (IVs) of exposure of interest and can be used to make causal inferences in complex relationships or to verify causal relationships indicated by observational studies [13]. Since single nucleotide polymorphisms (SNPs) are randomly assigned before birth and do not change due to disease, this is a similar approach to a randomized controlled trial compared to traditional observational studies, but MR is again less likely to be influenced by confounding factors and reverse causality. In addition, MR can distinguish independent causal effects of TH and SA from independent causal effects of sleep habits by selecting variants that are closely related to TH and SA but not to sleep habits.

Therefore, this paper aims to analyze the causal relationship between the five sleep habits of morningness, long sleep, short sleep, ease of getting up, and insomnia and brain cortical structure using MR in sleep characteristics. SNPs for sleep habituation tendencies are associated with certain TH and SA outcomes, suggesting a possible causal relationship between supportive sleep characteristics and TH and SA. To date, MR has not been used to assess the association between sleep characteristics and TH and SA outcomes.

## 2. Methods and Materials

Two-sample MR analyses were performed using pooled-level data to investigate the causal relationships between five sleep characteristics and TH and SA. The MR analyses were conducted based on following three assumptions. (1) Genetic variants are associated with five sleep habits: morningness, long sleep, short sleep, ease of getting up, and insomnia, which implies that genetic variants are associated with exposure; (2) IVs were not correlated with outcomes by confounders; and (3) IVs did not directly affect the brain cortical structure, but only possibly through exposure (Figure 1).

### 2.1. GWAS Summary Data for Cerebral Cortex Thickness and Surface Area

GWAS data of normal brain MRI related to cerebral cortex structure were discovered from the ENIGMA Consortium [6]. Of the 51,665 patients worldwide who used MRI to measure cortical TH and SA, all but about 6% were of European descent. 

Most importantly, our study used only meta results that included participants of European descent. These 34 regions were defined according to the Desikan–Killiany atlas, which establishes a rough subdivision of the cortex and identifies regional boundaries based on the anatomy of the cerebral gyrus [14]. Cerebral gyrus refers to a series of folds or creases located on the cerebral cortex, which increase the total surface area of the cerebral cortex, thus providing more neurons and functional regions. The 34 brain regions are divided as follows: bankssts, caudalanteriorcingulate, caudalmiddlefrontal, cuneus, entorhinal, frontalpole, fusiform, inferiorparietal, inferiortemporal insula, isthmuscingulate, lateraloccipital, lateralorbitofrontal, lingual, medialorbitofrontal, middletemporal, paracentral, parahippocampal, parsopercularis, parsorbitalis, parstriangularis, pericalcarine, postcentral, posteriorcingulate, precentral, precuneus, rostralanteriorcingulate, rostralmiddlefrontal, superiorfrontal, superiorparietal, superiortemporal, supramarginal, temporalpole, transversetemporal. We carried out MR Assessments of TH and SA in cortical regions of the brain, weighted estimates of TH and SA from early rise, long sleep, short sleep, ease of getting up, and insomnia, and obtained 68 outcomes. Data with globally weighted estimates showed SA and TH for particular areas of the brain as a whole, while data without globally weighted estimates only showed SA and TH for particular areas, excluding total brain. “Weighted estimates” refers to a statistical method involving assigning different weights to samples or genotypes to account for heterogeneity or imbalance within the samples. Adjusting the weights in this manner allows for a more accurate estimation of the association between genes and specific traits. So, without global weighted estimates were not considered in our study because the results obtained with global weighting were more accurate.

### 2.2. Genetic Variables Associated with Morningness, Long Sleep, Short Sleep, Ease of Getting Up, and Insomnia

For morningness, ease of getting up, and insomnia, in the analysis, we used the summary statistics of GWAS from Philip Jansen et al. [15] for morningness (215,945 cases and 129,607 controls), ease of getting up (317,949 cases and 68,000 controls), and insomnia (277,133 cases and 109,402 controls). We used the GWAS database of Dashti HS, et al. [16] to record short sleep (<7 h per night; *n* = 106,192 cases) and long sleep (≥9 h per night; *n* = 34,184 cases) relative to normal sleep (7–8 h per night; *n* = 305,742 controls) (https://sleep.hugeamp.org/downloads.html, accessed on 18 June 2023).

### 2.3. Selection of Genetic Instruments

Firstly, we MR randomized analysis using the corresponding GWAS database to examine the causal relationship between the five sleep characteristics and cortical SA and TH. To screen for IVs suitable for the five sleep-related traits, we extracted genome-wide meaningful SNPs associated with each exposure based on UK Biobank (*p* < 5 × 10^−8^), minor allele frequencies > 1% and not in linkage disequilibrium (LD) (R^2^ < 0.001, Kb = 10,000) to ensure independence between SNPs and eliminate linkage disequilibrium. Secondly, according to the requirements of the second MR hypothesis through the web site (www.phenoscanner.medschl.cam.ac.uk, accessed on 18 June 2023), we checked the genotype and phenotype information, so as to exclude any SNPs associated with these potential risk factors or confounding factors. Reported SNPs were associated with five sleep characteristics and potential confounders of cortical SA and TH, among them morningness (education, smoking, alcohol consumption, cardiovascular disease), ease of getting up (education, brain disease), insomnia (none), long sleep (none), and short sleep (psychosis) [17]. Third, moderate allele frequency palindromic SNPs were excluded. Finally, the palindromic SNPs were excluded.

We also calculated F values to ensure the strength of IVs in the MR analysis. Only SNPs with an F-statistic > 10 were considered reliable. The F of an instrumental SNP greater than 10 can reflect a strong correlation between SNP and exposure and reduce the influence of confounding factors.

Detailed information about the IVs used in the MR analysis is shown in Appendix A

### 2.4. Ethics

This study employed deidentified data from participant studies that had been authorized by an ethical standards council for human testing. This study did not require any further ethical approval.

### 2.5. Statistics

According to specific functions, the brain is divided into 34 different brain regions, and analysis is carried out based on specific brain regions as phenotypes. So, for the brain region-level analysis, considering 340 MR estimates, we also used Bonferroni-corrected *p* values (that is 0.05/340 = 1.47 × 10^−4^) to account for multiple testing. *p* values greater than Bonferroni-corrected *p* but less than 0.05 were considered suggestive of an association, and *p* values less than 0.01 were considered suggestive of a stronger association.

### 2.6. Mendelian Randomization Analysis

The Mendelian random instrument effect ratio was calculated by the coefficient ratio method (Wald), which is the ratio of the β coefficients of cortical thickness and surface area divided by the β coefficients of the corresponding sleep features, with standard errors calculated using the δ method [18]. This is a statistical technique used in regression analysis to determine the significance of a particular independent variable in explaining the dependent variable. For individual SNPs, MR instrument estimates were determined by the IVs ratio. In a preliminary analysis, inverse variance weighted (IVW) linear regression was used to estimate the causal effects of five sleep characteristics on cortical SA and TH, which yielded unbiased causal estimates in the absence of horizontal pleiotropy and heterogeneity [19]. In addition, four complementary MR methods, including MR-Egger, weighted median, simple mode, and weighted mode [20] were employed. The IVW random effects method is used as long as there is substantial heterogeneousness among IVs. The weighted median approach produced consistent results even when more than half of the IVs were invalid. MR-Egger regression was used to investigate the potential pleiotropic effects of selected IVs [21]. We also used the MR-pleiotropy residual sum and outlier (MR-PRESSO) method, which is for assessing whether an observed association between an exposure and an outcome is a causal relationship or if it is due to pleiotropic effects. This was used to reduce heterogeneity in the causal estimates by removing SNP-corrected horizontal pleiotropy that leads to outliers to a greater extent than expected (Nb distribution = 5000) [22]. Heterogeneity and pleiotropy provide evidence for the robustness of Mendelian randomization through the differentiation of instrumental variable variations and the specificity of causal direction. Meanwhile, MR-PRESSO identifies potential pleiotropic variables, making these three methods indispensable for sensitivity analyses in Mendelian randomization. All analyses were performed using the two-sample MR and MR-PRESSO packages (v1.0) in the R software (v0.5.6, R Foundation for Statistical Computing, Vienna, Austria) [19,22].

### 2.7. Sensitivity Analysis

To check the validity of the results, we performed heterogeneity, pleiotropy, and MR pleiotropy residual and outlier (MR-PRESSO) tests to determine significant horizontal pleiotropy. Heterogeneity in IVs was assessed using Cochran’s Q statistic, and *p* < 0.05 was considered heterogeneous [23]. Cochran’s Q test is a statistical test method used to assess heterogeneity present in a set of studies. It determines whether the variability in effect sizes across studies exceeds what would be expected by chance alone. A significant Q value (*p*-value < 0.05) indicates the presence of heterogeneity among the included studies. Forest plots and funnel plots were used to visualize heterogeneity [24]. As sensitive indicators of horizontal pleiotropy, intercepts, and *p*-values were calculated by MR-Egger regression analysis [25] and by IVW-RE leave-one-out analysis. Leave-one-out analysis is a method used in statistical modeling to assess the impact or importance of individual data points on the overall results, which is used to test whether causal estimates were driven by a single SNP [26]. 

MR-Egger regression examines whether there is a non-zero intercept in the regression of causal estimates on their corresponding precision. If the intercept is statistically significant (*p*-value < 0.05), it indicates the presence of pleiotropy, suggesting that instrumental variables may have a direct impact on the outcome, not solely through the exposure of interest. Moreover, sensitivity analyses were performed using significant SNPS (found only in Europe and the United States) to verify the association between these five sleep-related features and their significant or implied causal estimates between cortical thickness and surface area. The five sleep habits included morningness (59 SNPs), ease of getting up (35 SNPs), insomnia (11 SNPs), long sleep (5 SNPs), and short sleep (20 SNPs).

## 3. Results

In total, 59 indicator SNPs were screened for morningness gene prediction, 35 indicator SNPs for ease of getting up gene prediction, 11 SNPs for insomnia gene prediction, 5 SNPs for long sleep gene prediction, and 20 SNPs for short sleep gene prediction. The F-statistics of these genetic tools were all greater than the normal selection value of 10, indicating a strong correlation between IVs and the sleep traits they represent [27] Table 1.

Morningness (SA of cuneus, TH of frontal pole, SA of inferior parietal, SA of lateral occipital) and insomnia (TH of parahippocampal) correspond to cerebral cortex with the same SNP: rs77960. Morningness corresponding to four positives had the same SNP as ease of getting up (SA of lateral orbitofrontal): rs2653349, while there was no overlap between other sleep features. We conducted a comprehensive MR study from genetically predicted morningness, ease of getting up, insomnia, long sleep, and short sleep weighted on 34 functional brain gyri and identified several meaningful brain gyri influenced by the above five sleep characteristics (Figure 2). We present six scatter plots of results (Figure 3); the other results are shown in the Appendix A.

### 3.1. Association between Genetically Predicted Morningness and Cortical SA/TH

In the weighted analysis at the functional area level, the IVW model showed that morningness predicted by genetics significantly increased SA of cuneus (b = 32.63 mm, 95% CI: 10.35 mm to 54.90 mm, *p* = 0.004); weighted median was also of similar significance (b = 31.93, 95% CI: 0.69 mm to 63.16 mm, *p* = 0.045). MR-Egger, simple mode, and weighted mode methods yielded consistent directions of association.

The IVW model showed that genetically predicted morningness significantly reduced the TH of the frontal pole (b = −0.039 mm, 95% CI: −0.073 mm to −0.004mm, *p* = 0.025); MR-Egger, weighted median, simple mode, weighted mode methods yielded consistent direction of the association.

The IVW model showed a genetic prediction of morningness significantly reduced SA of the inferior parietal (b = −77.69, 95% CI: −150.50 mm to −4.89 mm, *p* = 0.036); MR-Egger, weighted median, simple mode, and weighted mode methods yielded a consistent direction of the association.

The IVW model showed a genetic prediction of morningness significantly increased SA of lateral occipital (b = 89.00, 95% CI: 26.66 mm to 151.34 mm, *p* = 0.005); the reliability of the causal estimates was high. The weighted median was similarly significant (b = 118.89, 95% CI: 35.38 mm to 202.40 mm, *p* = 0.005). The MR-Egger, simple mode, and weighted mode methods yielded consistent directions of association.

The heterogeneity assessed by Cochran’s Q-derived *p* value > 0.05 and pleiotropy using the MR-Egger intercept *p* value > 0.05. So, reliability of causal estimation was relatively high.

### 3.2. Association between Genetically Predicted Ease of Getting up and Cortical SA/TH

In the functional area level with overall weighted analysis, the IVW model showed that the genetic prediction of ease of getting up significantly increased the SA of lateral orbitofrontal (b = 57.65 mm, 95% CI: 3.26 mm to 112.04 mm, *p* = 0.038); the weighted median was similarly significant (b = 76.62, 95% CI: 2.33 mm to 150.92 mm, *p* = 0.043). The MR-Egger, simple mode, and weighted mode methods yielded consistent directions of association. The heterogeneity assessed by Cochran’s Q-derived *p* value > 0.05 and pleiotropy using the MR-Egger intercept *p* value > 0.05. So, reliability of causal estimation was relatively high.

### 3.3. Association between Genetically Predicted Insomnia and Cortical SA/TH

In functional area level containing overall weighted analysis, the IVW model showed a genetic prediction of insomnia significantly increased TH of parahippocampal (b = 0.002 mm, 95% CI: 0.0001 mm to 0.003 mm, *p* = 0.037); MR-Egger, weighted median, simple mode, and weighted mode methods yielded consistent association directions. The heterogeneity assessed by Cochran’s Q-derived *p* value > 0.05 and pleiotropy using the MR-Egger intercept *p* value > 0.05. So, reliability of causal estimation was relatively high.

### 3.4. Association between Genetically Predicted Long Sleep and Cortical SA/TH

In the functional area level with overall weighted analysis, the IVW model showed that genetically predicted long sleep significantly increased SA of isthmus cingulate (b = 274.93 mm, 95%CI: 63.03 mm to 486.84 mm *p* = 0.011); MR-Egger, weighted median, simple mode, and weighted mode methods yielded consistent directions of association.

The IVW model showed that genetically predicted long sleep significantly increased SA of parsopercularis (b = 409.03 mm, 95%CI: 0.02 mm to 73.42 mm, *p* = 0.017), and the Weighted median was similarly significant (b = 564.73, 95%CI: 162.76 mm to 966.69 mm, *p* = 0.006); MR-Egger, simple mode, and weighted mode methods yielded consistent directions of association. The heterogeneity assessed by Cochran’s Q-derived *p* value > 0.05 and pleiotropy using the MR-Egger intercept *p* value > 0.05. So, reliability of causal estimation was relatively high.

### 3.5. Association between Genetically Predicted Short Sleep and Cortical SA/TH

In the functional area level with overall weighted analysis, the IVW model showed that genetically predicted short sleep significantly reduced TH of the frontal pole (b = −0.18 mm, 95% CI: −0.33 mm to −0.03 mm, *p* = 0.019); MR-Egger, weighted median, and simple mode weighted mode methods yielded a consistent direction of the association.

The IVW model showed a genetic prediction of short sleep significantly reduced SA of the inferior parietal (b = −329.55 mm, 95% CI: −618.55 mm to −40.55 mm, *p* = 0.025); MR-Egger, weighted median, simple mode, and weighted mode methods yielded a consistent direction of the association.

The IVW model showed that genetic prediction of short sleep significantly increased SA of lateral occipital (b = 394.37, 95% CI: 107.89 mm to 680.85 mm, *p* = 0.007), and the weighted median was similarly significant (b = 423.90, 95% CI: 38.83 mm to 808.97 mm, *p* = 0.031); MR-Egger, simple mode, and weighted mode methods yielded consistent directions of association.

The IVW model showed that genetic prediction of short sleep significantly increased SA of middle temporal (b = 200.17, 95% CI: 12.97 mm to 387.37 mm, *p* = 0.036), with similar significance for weighted median (b = 292.71, 95% CI: 40.70 mm to 544.73 mm, *p* = 0.023); MR-Egger, simple mode, and weighted mode methods yielded consistent directions of association.

The IVW model showed that genetic prediction of short sleep significantly increased TH of middle temporal (b = 0.12, 95% CI: 0.05 mm to 0.20 mm, *p* = 0.002), and weighted median was similarly significant (b = 0.14, 95% CI: 1.03 mm to 1.28 mm, *p* = 0.011); MR-Egger, simple mode, and weighted mode methods yielded consistent directions of association. 

The IVW model showed that genetic prediction of short sleep significantly reduced the TH of paracentral (b = −0.11, 95% CI: −0.19 mm to −0.03 mm, *p* = 0.006); MR-Egger, weighted median simple mode, and weighted mode methods yielded a consistent direction of the association.

The IVW model showed a genetic prediction of short sleep significantly reduced TH of parahippocampal (b = −0.25, 95% CI: −0.42 mm to −0.07 mm, *p* = 0.006); MR-Egger, weighted median simple mode, and weighted mode methods yielded a consistent correlation direction. 

IVW model showed a genetic prediction of short sleep significantly increased TH of superior temporal (b = 0.09, 95% CI: 0.02 mm to 0.16 mm, *p* = 0.013), with similar significance for weighted median (b = 0.15, 95%CI: 0.05mm to 0.25 mm, *p* = 0.004); MR-Egger, simple mode, and weighted mode methods yielded consistent directions of association. 

The heterogeneity assessed by Cochran’s Q-derived *p* value > 0.05 and pleiotropy using the MR-Egger intercept *p* value > 0.05. So, reliability of causal estimation was relatively high (Appendix A).

For stronger association and association estimates, horizontal multiplicity was assessed using Cochran’s q-test, MR-Egger intercept test, leave-one-out analysis, and funnel plot. All *p*-values for the MR-Egger intercept test were > 0.05 (Appendix A), indicating that horizontal multiplicity did not exist. In the Appendix A, leave-one-out analyses, and funnel plot in Appendix A show that these estimates are not affected by a single SNP. The estimated value meets the requirement that all *p* values of Cochran’s Q are > 0.05. 

## 4. Discussion

To our knowledge, this is the first large-scale MR analysis to comprehensively determine the causal relationships between morningness, ease of getting up, insomnia, long sleep, and short sleep, and cortical structure (cortical SA and cortical TH in 34 brain regions). The radial unit hypothesis [28] posits that the expansion of cortical SA is driven by the proliferation of these neural progenitor cells, whereas thickness is determined by the number of their neurogenic divisions. In other words, the cortical surface area is related to the number of cortical columns and can reflect the folding of the cortex, whereas cortical thickness, which is the distance between the cortical surface and the white matter, reflects the thickness of the cortical gray matter and is related to the number of cells within the cortical columns. In addition, volume is the product of thickness and surface area, and an increase in cortical volume can be caused by a thickening of cortical thickness and/or an increase in surface area, so the two independent parameters are more relevant than the quantification of volume. Of course, the measurement of volume is also important because changes in cortical volume may not always correspond to changes in surface area and thickness (changes in surface area and thickness may be non-differential).

In the current MR study, we systematically evaluated the contingent relationship between five genetically predicted sleep traits and cortical structures. Our findings suggest that sleep traits can influence the cerebral cortex and support the findings of earlier observational studies [29] that suggest pathophysiological interactions between sleep traits and brain function, thus underscoring the existence of a reciprocal regulatory function between sleep and the cerebral cortex.

The cerebral cortex is the highest-level center for the regulation of somatic movements and is composed of three parts: primary sensory areas, primary motor areas, and joint areas. The human cerebral cortex has about 14 billion nerve cells and covers an area of about 2200 square centimeters, accounting for about 80% of the entire brain volume. It is responsible for many complex systems of the brain, including thinking, language, perception, etc.

The quality of sleep may be related to the cerebral cortex, and how long and how deep is perhaps regulated by the cerebral cortex responsible. Moreover, in an Oxford study, it was indicated that after deactivating neurons in layer five of the neocortical layers of the cerebral cortex, mice woke up for 3 h more per day while not needing deeper sleep to compensate [30]. This suggests that the cerebral cortex also plays a role in the regulation of sleep and waking.

Brain function can be restored during sleep through repair and cleaning mechanisms. The “lymphatic hypothesis” mechanism suggests that lymphatic flow is accelerated during NREM sleep to bring out excess metabolites and help brain repair [31,32]. Sleep deprivation is highly damaging to the brain, and if it occurs, more time is needed to repair the brain, and sleep deprivation can lead to altered levels of short-chain fatty acid metabolism in the human gut, which can affect neuroinflammation levels [33].

Patients with sleep disorders are at relatively high risk for cognitive impairment and neuropsychiatric disorders, and these effects include altered cortical structure and altered brain function/neuropsychiatric disorders. Significant correlations were found between different psychiatric disorders and multiple sleep indicators, and the greatest correlation was found between depressive disorders and sleep efficiency [34]. The potential association between sleep disorders and cortical structures or a range of neuropsychiatry is not fully elucidated. Brain cortex structure, especially cortical thickness, is considered a neuroimaging biomarker for predicting cognitive decline. In addition, alterations in brain structure can provide insights [35]. When MR is used to reveal the causal relationship between sleep and neuropsychiatric disorders, it is more objective to reveal the causal relationship between sleep characteristics and structural alterations. Inspired by Krone et al. [30], we performed an MR examination to explore the effect of sleep characteristics on cortical structure. We plan to investigate how these changes affect brain function and neuropsychiatry, as well as carrying out in-depth research for improvement in surveillance and early detection for patients at high risk of related diseases and provide necessary programs for the prevention and medication of cognitive impairment and neurological and psychiatric disorders in patients with sleep disorders.

Our principal work found that morningness was causally associated with increased SA in the cuneus and lateral occipital regions; meanwhile, short sleep was associated with increased SA in the lateral occipital region, increased TH in the middle temporal gyrus, and decreased TH in the paracentral lobule and hippocampal parabasal gyrus cortex. Notably, there is evidence that the precuneus, middle temporal gyrus (including hippocampus, parahippocampal gyrus, and internal olfactory cortex, among others), cingulate gyrus, and prefrontal lobe play an important role in situational memory [36]. Precuneus are closely connected to limbic subcortical structures (hippocampus, amygdala, and thalamus), regulating higher cognitive functions, including memory retrieval, attention, and conscious perception [37]. Moreover, increased left cuneate volume is significantly associated with enhanced working memory [38] and increased cuneate TH is associated with reduced risk of AD, presumably cuneate atrophy occurs prior to situational memory loss and it may contribute to the diagnosis of risk of AD [39]. Furthermore, the lateral occipital cortex has complex functions, including object recognition, face recognition, and motion perception [40]. It co-integrates and analyzes visual information with auditory and other sensory information and is linked to brain regions that facilitate speech and other executive functions and is associated with visual memory. Therefore, we venture to infer that the morningness makes the increased surface area of the cuneus and lateral occipital regions helpful in enhancing working learning memory, which can also be expected in future studies.

Surprisingly, both early rising and short sleep increased the SA of the lateral occipital cortex. An observational study by Sasabayashi et al. found that patients with an at-risk mental state (ARMS) exhibited elevated cortical TH in the right occipital region [41]; however, their study found that it may be more the right occipital cortical TH than the surface area that is associated with ARMS. Moreover, patients with Parkinson’s disease with stable normal cognition experienced a progression of cortical thinning mainly in the occipital and parietal regions [42]. This suggests that the underlying mechanisms of sleep habits and complex alterations in the lateral occipital or even the entire occipital cortex and other neuropsychiatric disorders deserve further investigation. Another clinical study showed that individuals with ARMS with poor functional outcome exhibited a significant reduction in right paracentral lobule cortical TH compared to individuals with ARMS with the good functional outcome [41], while our study demonstrated that short sleep characteristics can also lead to reduced paracentral lobule cortical TH. Short sleep has also been associated with a reduction in paracentral hippocampal cortex TH, and cortical TH in the left paracentral hippocampal cortex is associated with central hearing and cognitive abilities in old age and is negatively correlated with a hearing threshold [43] (the less cortical TH, the worse hearing). 

Incidentally, decreased TH in the temporal lobe was associated with first episode psychosis (FEP), a brain region critical to deliberation processes, autobiographical memory, and constructing other people’s perspectives, and whose dysfunction is associated with the core symptoms of psychosis [35]. And Parkinson’s disease characterized by frequent impulse onset has been observed in patients with decreased gray matter volume in the bilateral superior and middle temporal gyri [44]. In contrast, our study found that short sleep causes an increase in cortical TH in this area, which is different from the effects exerted by short sleep in other brain regions. Our estimates suggest that short sleep leads to increased SA in the lateral occipital area, increased TH in the middle temporal gyrus, and decreased TH in the paracentral lobule and hippocampal paracentral gyrus cortex, but whether these changes contribute to these neuropsychiatric disorders by affecting SA and TH remains to be investigated. TH changes have also been reported in dystonia [45], vocal tremor [46], Parkinson’s disease, depression, schizophrenia, attention deficit hyperactivity disorder, and insomnia [47].

Despite only six estimates with *p* < 0.01, other IVW *p* values < 0.05 should be treated with caution. It is noteworthy that some estimates differ from logical expectations. Decreased thickness of the frontopolar cortex and decreased SA of the inferior parietal lobule were associated with morningness and short sleep, and increased TH of the superior temporal gyrus was associated with short sleep. Increased TH of the parahippocampal gyrus was associated with insomnia. Increased SA of the lateral orbitofrontal was associated with ease of getting up. Increased SA of the isthmus of the cingulate gyrus and the insula of the inferior frontal gyrus was associated with long sleep.

The frontal pole is an area essential for higher cognitive functions, is highly connected to many other brain regions, and symptoms of schizophrenia are closely associated with dysfunction in this key brain region [48]. Patients with schizophrenia have a smaller total cortical SA and thinner mean TH, with the greatest effects in the frontal and temporal lobes [49]. However, in our study, both morningness and short sleep phenotypes resulted in a tendency to reduce cortical thickness in the frontopolar (anterior part of the frontal lobe), and short sleep showed a tendency to make them increase on the surface area and thickness of the middle temporal gyrus, as well as on the thickness of the superior temporal gyrus. The complex causality of this needs to be further investigated. Unexpectedly, the causal estimation of our findings of insomnia and possible increased thickness of the parahippocampal gyrus differed from logical expectations. McLachlan et al. [50] reported that reduced parahippocampal volume in AD was associated with the formation of misidentification delusions. In older adults with biaxial affective disorder, there is extensive cortical thinning in which the thickness and volume of the parahippocampal cortex are reduced [51]. Again, we did not detect heterogeneity and pleiotropy, and the mechanisms of this need further investigation. The sleep characteristics of ease of getting up and the increased SA of the lateral orbitofrontal cortex may have a causal relationship. The orbitofrontal cortex (OFC) is part of the prefrontal cortex that is mainly involved in cognitive functions, and individuals rely on the OFC to make choices or avoid risks [52,53], and damage to the orbitofrontal cortex can lead to severe emotional loss. It has been shown that structural and functional impairment of the orbitofrontal cortex is present in the early stages of schizophrenia [54], where a decrease in OFC gray matter volume, cortical TH, and SA were found [55]. Therefore, changes in OFC volumetric gray matter correlate with the severity of negative and cognitive deficits in schizophrenia. There is a causal relationship between long sleep and increased SA of the cingulate gyrus. Observational studies have found that patients with concomitant depression show significant increases in the SA of the left cingulate isthmus as well as in gray matter volume in the same brain region and that these abnormally enlarged brain regions predict their clinical depressive symptoms. However, some studies have reported that depression is associated with a decrease in cingulate gyrus volume, TH, and SA, while others have demonstrated an increase in morphometry [56]. These changes in the isthmic cingulate may indirectly reflect dysfunctional emotional responses and an impaired ability to process emotional stimuli derived from situational memory [57]. It is suggested that structural abnormalities in the cingulate gyrus may be an important neurobiological marker of depressive symptoms associated with schizophrenia, with important implications for the independent diagnostic identification of depressive disorders associated with schizophrenia [58]. Long sleep may be causally related to the increased SA of pars opercularis. The pars opercularis is an important language site [59], and alterations have been reported in patients with multiple sclerosis [60]. Inhof et al. also reported that the volume of the pars opercularis was associated with Internet addiction [61]. Curley et al. found in an observational study that pars opercularis SA were associated with better motor inhibition performance [62]. Thus, increased pars opercularis area may improve attention deficit, hyperactivity, and executive ability problems. 

To our knowledge, this is the first study using MR to address the causal relationship between sleep habits and cortical structures. Our survey provides new evidence for the effect of sleep habits on cortical changes, which in turn affect brain function. Still, this work has a drawback. Firstly, the sample was mostly European; therefore, the effect of sleep characteristics on cortical structural changes is unknown except in European populations. Secondly, our summary describe alterations in cortical structure in only five sleep trait populations, but the underlying mechanisms deserve further investigation. Thirdly, sleep disorder is approximately 1.5 times more common in women than in men [63]. However, there is still limited literature exploring the gender differences in the impact of insomnia on brain anatomical structures. In addition, previous studies exploring the impact of insomnia on brain structure have also considered the influence of covariates such as gender and age [64]. MR is a methodology that employs genetic variants as instrumental variables to investigate causal relationships between exposure factors and outcomes. Gender and age, considered as fundamental individual traits, are generally not regarded as primary factors. Our primary objective in this study is to investigate the effect of overall sleep status on the cortex among the general population. Future advancements GWAS will enable us to probe the effects of sleep on diverse age groups and genders more extensively. Fourthly, our sample size is moderate, but it is currently the largest among GWAS data. Additionally, the collection of phenotype data is based on self-report, which may introduce subjectivity. However, the impact of subjective bias decreases as the sample size increases. Lastly, we use the two-sample MR method, and in the future, we can employ bidirectional MR to validate the robustness of this causal relationship.

## 5. Conclusions

Our data provide a bridge to explore the link between sleep characteristics and other neuropsychiatric disorders, with a particular focus on more sleep and the specific brain gyrus of short sleep. Efforts are being made to discover the mechanism of the link between short sleep and brain diseases, and to research new therapeutic approaches to better care for neuropsychiatric disorders caused by short sleep. This is the first comprehensive MR analysis to reveal the association between morningness, ease of getting up, insomnia, long sleep, and short sleep, and cortical structures. Our estimates suggest that morningness sleep characteristics lead to increased surface area in the cuneus and lateral occipital regions; meanwhile, short sleep leads to increased surface area in the lateral occipital region and increased thickness in the middle temporal gyrus, while leading to decreased cortical thickness in the paracentral lobule and hippocampal parabrachial gyrus. Brain MRI has the potential to be used for the early diagnosis of neuropsychiatric disorders in short-sleeping populations. The mechanism of the relationship between sleep characteristics and altered brain function needs to be further investigated.

## Figures and Tables

**Figure 1 biomedicines-11-02296-f001:**
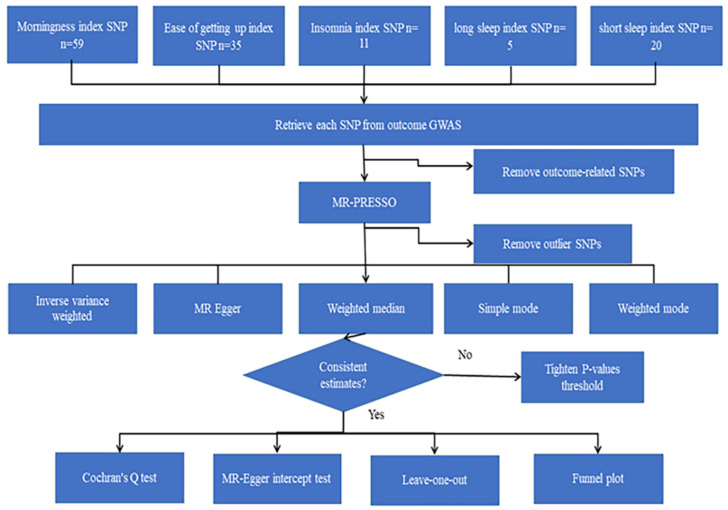
The flowchart of the two-sample MR analyses. Single nucleotide polymorphism (SNPs), Genome-wide association study (GWAS), MR-pleiotropy residual sum, and outlier (MR-PRESSO). “n” refers to the number of instrumental variables.

**Figure 2 biomedicines-11-02296-f002:**
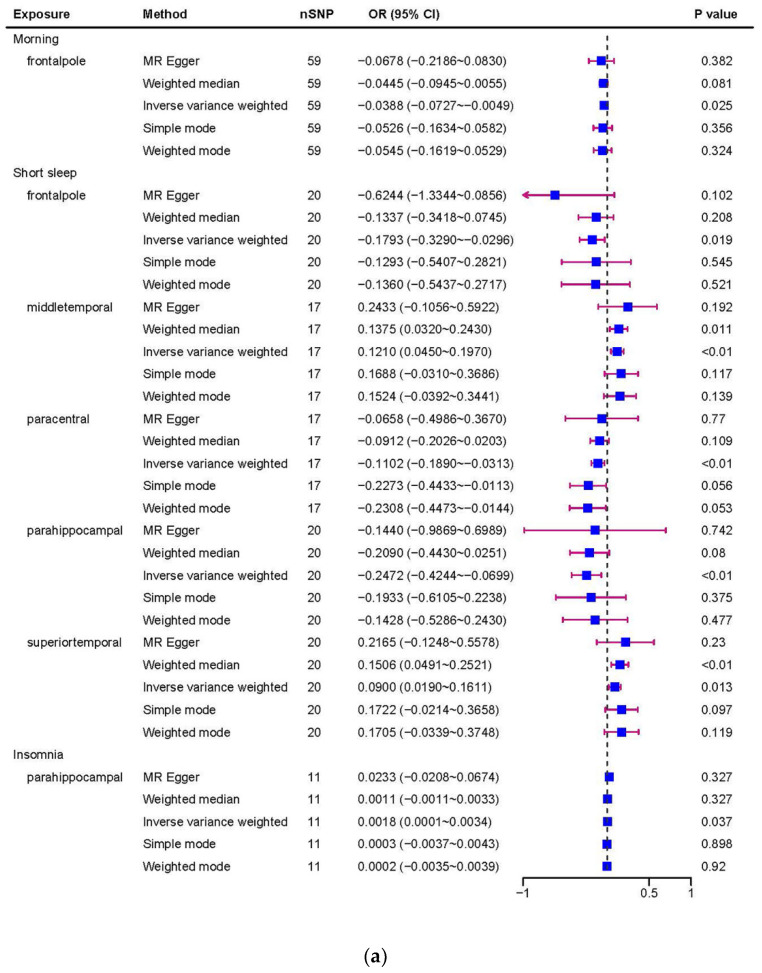
(**a**) The association of sleep traits and cortical thickness. (**b**) The association of sleep traits and cortical surficial area.

**Figure 3 biomedicines-11-02296-f003:**
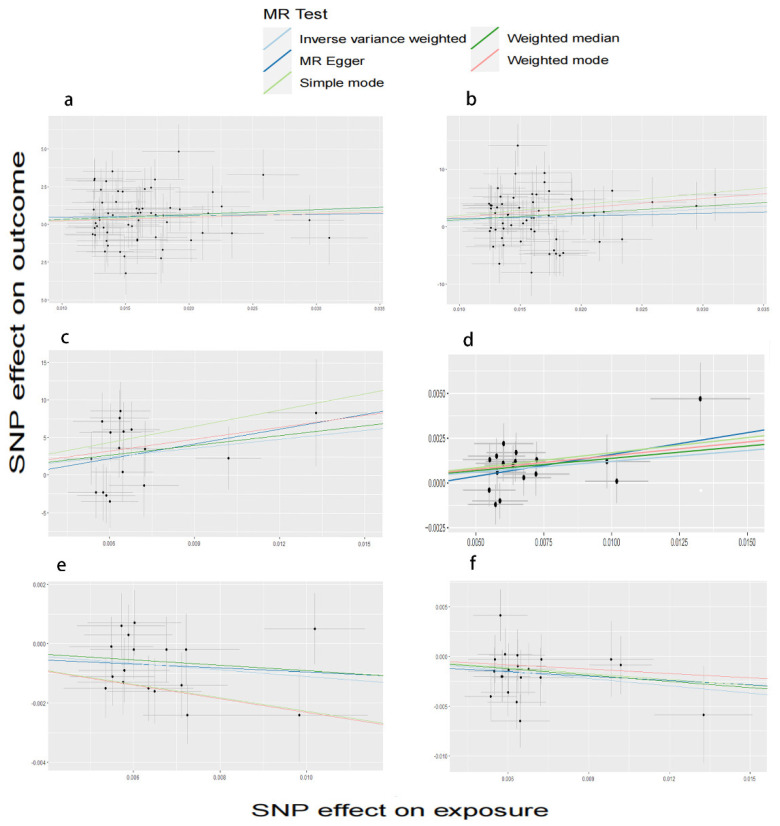
IVW estimates of stronger association results from morningness, long sleep, short sleep, ease of getting up, and insomnia on cortical SA and TH with global weighting. (**a**) Scatter plots from genetically predicted morningness on SA of the cuneus; (**b**) scatter plots from genetically predicted morningness on SA of the lateral occipital; (**c**) scatter plots from genetically predicted short sleep on SA of the lateral occipital; (**d**) scatter plots from genetically predicted short sleep on TH of the middle temporal; (**e**) scatter plots from genetically predicted short sleep on TH of paracenreal; and (**f**) scatter plots from genetically predicted short sleep on TH of parahippocampal.

**Table 1 biomedicines-11-02296-t001:** The association of sleep traits and cortical surficial area. SA: surficial area; TH: thickness.

Exposure	Outcome	IVW-Derived *p* Value	Beta (95% Confidence Intervals)	Cochran’s Q-Derived *p* Value	MR-Egger Intercept Derived *p* Value
Morningness	SA of cuneus	0.004	32.63 (10.35, 54.90)	0.14	0.69
	TH of frontal pole	0.025	−0.04 (−0.07, −0.005)	0.76	0.70
	SA of inferior parietal	0.036	−77.69 (−150.50, −4.89)	0.11	0.86
	SA of lateral occipital	0.005	89.00 (26.66, 151.34)	0.083	0.70
Ease of getting up	SA of lateral orbitofrontal	0.038	57.65 (3.26, 112.04)	0.22	0.62
Insomnia	TH of parahippocampal	0.037	0.002 (0.0001, 0.003)	0.66	0.36
Long sleep	SA of isthmus cingulate	0.011	274.93 (63.03, 486.84)	0.60	0.90
	SA of parsopercularis	0.017	409.03 (73.42, 744.64)	0.59	0.79
Short sleep	TH of frontal pole	0.019	−0.18(−0.33, −0.03)	0.61	0.22
	SA of inferior parietal	0.025	−329.55 (−618.55, −40.55)	0.59	0.25
	SA of lateral occipital	0.007	394.37 (107.89, 680.85)	0.34	0.73
	SA of middle temporal	0.036	200.17 (12.97, 387.37)	0.71	0.38
	TH of middle temporal	0.002	0.12 (0.05, 0.20)	0.31	0.76
	TH of paracentral	0.006	−0.11 (−0.19, −0.03)	0.69	0.84
	TH of parahippocampal	0.006	−0.25 (−0.42, −0.07)	0.81	0.81
	TH of superior temporal	0.013	0.09 (0.02, 0.16)	0.42	0.47

## Data Availability

Summary data from GWAS for cortical thickness and surficial area are available at https://enigma.ini.usc.edu/downloads (accessed on 18 June 2023); summary data for morningness, ease of getting up early, and insomnia are from https://ctg.cncr.nl/software/summary_statistics (accessed on 18 June 2023); summary data for long sleep and short sleep are from https://sleep.hugeamp.org/downloads.html (accessed on 18 June 2023); the R source code for processing the data is also included in the Appendix A, including the formula for calculating F.

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
