# Peer review of "Sleep Traits Causally Affect the Brain Cortical Structure: A Mendelian Randomization Study"

_biomedicines, 2023, doi:10.3390/biomedicines11082296_

Round 1

Reviewer 1 Report

This is a very interesting and relevant study that advances our knowledge about the alterations in the brain structures and sleep characteristics. The work is original and well-performed.

Minor comments:

1.       In abbreviations, please include SNPs, GWAS.

2.       Text in Figure A2 plots (plot axis and legends) is small and difficult to read, please enlarge.

3.       For Figure A2, since all the plots (a-f) share the same color legends, consider having it only once at the top of figure. Please add a short title above each plot so it is easy to interpret without having to read through the text. Also add R-squared values for each plot.

4.       Figure A3, cortical is misspelled as cortica.

5.       Please cite and include discussion about how gender and age affect affects brain structures (SA, TH etc), and how your results and interpretation accounted for these changes (or how they could affect interpreting these results).

6.       Text in the supplementary figures (axis and plot legends) are too small to read, please enlarge. Similar to comment (3) above, if they share the same plot legend throughout the figure, consider only have it once.

Author Response

Thank you very much for your careful review and constructive suggestions with regard to our manuscript. Meanwhile, I am also very grateful to you for the specific modification strategy.I have made comprehensive and detailed changes according to your suggestions

  1. In abbreviations, please include SNPs, GWAS.
  2. Text in Figure A2 plots (plot axis and legends) is small and difficult to read, please enlarge.
  3. For Figure A2, since all the plots (a-f) share the same color legends, consider having it only once at the top of figure. Please add a short title above each plot so it is easy to interpret without having to read through the text. Also add R-squared values for each plot.
  4. Text in the supplementary figures (axis and plot legends) are too small to read, please enlarge. Similar to comment (3) above, if they share the same plot legend throughout the figure, consider only have it once.

We are very sorry for our negligence of standardized expression of abbreviations,considering the Reviewer’s suggestion, we used SNP and GWAS and we revised it strictly according to the suggestion in the abstract and manuscript.We simultaneously improved the images based on the modification requirements of the pictures.

  1. Figure A3, cortical is misspelled as cortica.

We are very sorry for our negligence of correct expression of abbreviations,considering the Reviewer’s suggestion, we used cortica strictly

  1. Please cite and include discussion about how gender and age affect affects brain structures (SA, TH etc), and how your results and interpretation accounted for these changes (or how they could affect interpreting these results).

It is really true as Reviewer suggested, we accept your comment and have added it at the last paragraph of the discussion, the one that points out the purpose.

“Sleep disorder is approximately 1.5 times more common in women than in men.However, there is still limited literature exploring the gender differences in the impact of insomnia on brain anatomical structures.In addition, previous studies exploring the impact of insomnia on brain structure have also considered the influence of covariates such as gender and age. MR is a methodology that employs genetic variants as instrumental variables to investigate causal relationships between exposure factors and outcomes.Gender and age, considered as fundamental individual traits, are generally not regarded as primary factors .our primary objective in this study is to investigate the effect of overall sleep status on the cortex among the general population.Future advancements GWAS will enable us to probe the effects of sleep on diverse age groups and genders more extensively”.

Reviewer 2 Report

This is an interesting study examining sleep traits and brain cortical structure using Mendelian randomization. I have several comments to improve the manuscript further:

1. In the first paragraph, it would be clearer if the authors specify the units for sleep duration. For example, "< 6 hours of sleep per night" instead of just "< 6 hours."

2. In the method section, consider rephrasing "The MR analyses were consistent with three hypotheses" to "The MR analyses were conducted based on three assumptions" to better convey that these assumptions underpin the MR analysis.

3. In the first paragraph of the third section, it was stated that GWAS data related to brain cortical structures were obtained from the ENIGMA Consortium. It would be helpful to provide a brief explanation of what GWAS stands for (Genome-Wide Association Study) for readers who may not be familiar with the term.

4. In the sentence 105, consider clarifying what "cerebral gyrus marked between the depths of the cerebral sulci" means for better understanding.

5. The phrase "with the weighted estimates, yielding 68 outcomes" is unclear. Please clarify what is meant by "weighted estimates" and "68 outcomes."

6. It would be helpful to explain why the results obtained with global weighting were considered more accurate compared to those without global weighted estimates.

7. Please clarify what "analysis we used the latest summary statistics of GWAS" refers to. Are these summary statistics from a specific study or database?

8. "SNPs were excluded if they were associated with potential risk factors" may benefit from further clarification. Are these risk factors related to sleep characteristics or the outcomes of cortical SA and TH?

9. Sentence 141 mentions excluding GWAS-unavailable SNPs. Please clarify what is meant by "GWAS-unavailable" to ensure understanding.

10. In sentence 144, consider mentioning why an F-statistic threshold of >10 was chosen to determine the strength of IVs in the MR analysis.

11. It would be helpful to explain what "region-level analysis" refers to for better comprehension.

12. The term "coefficient ratio method (Wald)" can be further explained or referenced to provide more context.

13. Please provide more context or explanation of the MR-PRESSO method and its purpose.

14. Please specify the versions of the "Two Sample MR" and "MR-PRESSO" packages used in the R software for transparency.

15. I agree that the sensitivity analysis is important. it would be beneficial to the readers to explain why heterogeneity, pleiotropy, and MR-PRESSO tests were chosen as sensitivity analyses and how they contribute to assessing the robustness of the results.

16. "No pleiotropy or heterogeneity was found. The reliability of the causal estimates was high," consider providing additional information on how pleiotropy and heterogeneity were assessed to support these claims.

17. where it discusses horizontal multiplicity and the tests conducted, consider providing more context or explanation for readers who may not be familiar with these statistical tests. Briefly explain Cochran's q-test, MR-Egger intercept test, and leave-one-out analysis.

18. In sentence 309, specify which estimates were not violated (e.g., estimates of causal effects, estimates of heterogeneity).

19. Several limitations should be elaborated such as sample population, any potential limitation regarding Mendelian randomization, measurement issues (especially related to sleep traits) and potential heterogeneity,

Author Response

Reviwer 2

Thank you very much for your careful review and constructive suggestions with regard to our manuscript. Meanwhile, I am also very grateful to you for the specific modification strategy.I have made comprehensive and detailed changes according to your suggestions

  1. In the first paragraph, it would be clearer if the authors specify the units for sleep duration. For example, "< 6 hours of sleep per night" instead of just "< 6 hours."
  2. In the method section, consider rephrasing "The MR analyses were consistent with three hypotheses" to "The MR analyses were conducted based on three assumptions" to better convey that these assumptions underpin the MR analysis.

Thank you for your advice.We have re-written this part according to the Reviewer’s suggestion.

  1. In the first paragraph of the third section, it was stated that GWAS data related to brain cortical structures were obtained from the ENIGMA Consortium. It would be helpful to provide a brief explanation of what GWAS stands for (Genome-Wide Association Study) for readers who may not be familiar with the term.

It is really true as Reviewer suggested, we accept your comment and have added it at the last paragraph of the discussion, the one that points out the purpose.The content is shown below.

“GWAS (Genome-Wide Association Study) is a method that investigates the associations between the human genome and specific traits, such as disease risk. It involves analyzing genomic data from a large number of individuals to identify the relationship between DNA variations and specific traits.”

  1. In the sentence 105, consider clarifying what "cerebral gyrus marked between the depths of the cerebral sulci" means for better understanding.

We have clarified that“cerebral gyrus refers to a series of folds or creases located on the cerebral cortex, which increase the total surface area of the cerebral cortex, thus providing more neurons and functional regions”

  1. The phrase "with the weighted estimates, yielding 68 outcomes" is unclear. Please clarify what is meant by "weighted estimates" and "68 outcomes."
  2. It would be helpful to explain why the results obtained with global weighting were considered more accurate compared to those without global weighted estimates.

We further explained it that “weighted estimates" refers to a statistical method involving assigning different weights to samples or genotypes to account for heterogeneity or imbalance within the samples. Adjusting the weights in this manner allows for a more accurate estimation of the association between genes and specific traits.

  1. Please clarify what "analysis we used the latest summary statistics of GWAS" refers to. Are these summary statistics from a specific study or database?

Thank you for your question.Yes, the GWAS data we used is from the largest publicly available database.

  1. "SNPs were excluded if they were associated with potential risk factors" may benefit from further clarification. Are these risk factors related to sleep characteristics or the outcomes of cortical SA and TH?

Thank you for your question.These SNPs refer to confounding factors that can cause changes in brain structure.

  1. Sentence 141 mentions excluding GWAS-unavailable SNPs. Please clarify what is meant by "GWAS-unavailable" to ensure understanding.

Thank you for your question.palindromic SNP refers to the single nucleotide polymorphism (SNP) with symmetric base pairing in the DNA sequence.If palindromic SNPs are retained in the design and analysis of experiments, it may lead to the phenomenon of DNA fragment concatenation and reverse complementarity, resulting in misinterpretation of experimental results.

10.In sentence 144, consider mentioning why an F-statistic threshold of >10 was chosen to determine the strength of IVs in the MR analysis.

Thank you for your question.Mendelian randomization with an instrumental variable F greater than 10 can reflect a strong correlation between SNP and exposure and reduce the influence of confounding factors.

11.It would be helpful to explain what "region-level analysis

It is really true as Reviewer suggested, we accept your comment and have added it this section, the one that points out the purpose.According to specific functions, the brain is divided into 34 different brain regions, and analysis is carried out based on specific brain regions as phenotypes.

  1. The term "coefficient ratio method (Wald)" can be further explained or referenced to provide more context.

It is really true as Reviewer suggested, we accept your comment and have added it this section, “the one that points out the purpose,the coefficient ratio method, also known as the Wald method, is a statistical technique used in regression analysis to determine the significance of a particular independent variable in explaining the dependent variable”.

  1. Please provide more context or explanation of the MR-PRESSO method and its purpose.

It is really true as Reviewer suggested, we accept your comment and have added it this section, “the one that points out the purpose,the MR-PRESSO (Mendelian Randomization Pleiotropy RESidual Sum and Outlier) method is a statistical technique used in the field of genetics and epidemiology. Its purpose is to assess whether an observed association between an exposure and an outcome is a causal relationship or if it is due to pleiotropic effects.”

  1. Please specify the versions of the "Two Sample MR" and "MR-PRESSO" packages used in the R software for transparency.

Thank you for your question.The version of the software TwoSampleMR is 0.5.6 and MR-PRESSO is 1.0

  1. I agree that the sensitivity analysis is important. it would be beneficial to the readers to explain why heterogeneity, pleiotropy, and MR-PRESSO tests were chosen as sensitivity analyses and how they contribute to assessing the robustness of the results.

It is really true as Reviewer suggested, we accept your comment and have added it this section,

Heterogeneity” refers to the variability in causal estimates across different genetic variants used as instrumental variables;“MR-PRESSO tests” are specifically designed to detect and correct for pleiotropic effects and outliers in MR analysis. These tests help identify genetic variants that significantly deviate from the majority of variants in their estimated causal effect; “Pleiotropy”: In MR analysis, pleiotropy can be problematic as it violates the assumption that the genetic variant used as an instrument only affects the outcome indirectly via the exposure of interest. Heterogeneity and pleiotropy provide evidence for the robustness of Mendelian randomization through the differentiation of instrumental variable variations and the specificity of causal direction. Meanwhile, MR-PRESSO identifies potential pleiotropic variables, making these three methods indispensable for sensitivity analyses in Mendelian randomization.

  1. "No pleiotropy or heterogeneity was found. The reliability of the causal estimates was high," consider providing additional information on how pleiotropy and heterogeneity were assessed to support these claims.

It is really true as Reviewer suggested, we accept your comment and have added it this section,firstly, we assessed heterogeneity using Cochran's Q-derived P value (>0.05) and pleiotropy using the MR-Egger intercept P value (>0.05). Additionally, we performed MR-PRESSO analysis to identify and remove any outlier SNPs.This makes the reliability of causal estimation relatively high.”

  1. where it discusses horizontal multiplicity and the tests conducted, consider providing more context or explanation for readers who may not be familiar with these statistical tests. Briefly explain Cochran's q-test, MR-Egger intercept test, and leave-one-out analysis.

It is really true as Reviewer suggested, we accept your comment and have added it this section,the MR-Egger intercept test is a statistical test method used in Mendelian randomization (MR) studies to assess pleiotropy (a form of horizontal pleiotropy). It examines whether there is a non-zero intercept in the regression of causal estimates on their corresponding precision. If the intercept is statistically significant (p-value < 0.05), it indicates the presence of pleiotropy, suggesting that instrumental variables may have a direct impact on the outcome, not solely through the exposure of interest.Cochran's Q test is a statistical test method used to assess heterogeneity present in a set of studies. It determines whether the variability in effect sizes across studies exceeds what would be expected by chance alone. A significant Q value (p-value < 0.05) indicates the presence of heterogeneity among the included studies.Leave-one-out analysis is a method used in statistical modeling to assess the impact or importance of individual data points on the overall results.

  1. In sentence 309, specify which estimates were not violated (e.g., estimates of causal effects, estimates of heterogeneity).]

Thank you fou your advice.We have clarified the sentence to “In the supplemental material, leave-one-out analyses and funnel plot in Figure S1-14 show that these estimates are not affected by a single SNP. The estimated value meets the requirement that all p values of Cochran's Q are >0.05,so the sensitive analysis of this study can identified its robustness” .

  1. Several limitations should be elaborated such as sample population, any potential limitation regarding Mendelian randomization, measurement issues (especially related to sleep traits) and potential heterogeneity,

Our sample size is moderate, but it is currently the largest among GWAS data. Additionally, the collection of phenotype data is based on self-report, which may introduce subjectivity. However, the impact of subjective bias decreases as the sample size increases. Lastly, we used the two-sample MR method, and in the future, we can employ bidirectional MR to validate the robustness of this causal relationship.

Reviewer 3 Report

biomedicines-2436145: “Sleep traits causally affect the brain cortical structure: A Mendelian randomization study”.

In this study, the authors analyze associations between some abnormal sleep characteristics and differential changes in the cortical tissue parameters hypothesizing that this phenomenon might be involved in mechanisms of some neurological disorders. The material is described successively and conclusions are partially supported by obtained data.

Remarks/recommendations:

  1. Absolutely unclear through the text, whether participants were all healthy persons or not;
  2. Abstract should be simplified by removing of the numerical data and “(SA)”;

In line 23, “TH” should be replaced by “cortical thickness (TH)”;

  1. In line 29, “…parahippocampal brain areas….” and “…the middle temporal zone…”;
  2. In lines 28-30, the statement is incorrect and should be rewritten;
  3. In line 49, “…decline. Measurements of brain cortical surface area (SA) and its TH…”;
  4. In line 50 and everywhere below, references in brackets should be included separately;
  5. In line 60, “…in the carriers (10). “;
  6. In line 67, “…require...” ;
  7. In lines 74 and 76, “IVs” and “SNPs”, respectively, should be open;
  8. In line 80, just “SA”;
  9. In line 93, “exposure” should be clarified;
  10. Figure 1, “SNP”, “GWAS”, “MR-PRESSO” and “n=” should be described in the legend;
  11. In line 103,104, “These 34 regions…” needs to be clarified;
  12. In line 108, “…of TH and SA…” should be removed;
  13. In lines 121-123, “We used the GWAS database of Dashti HS, et al. (17) for short sleep (< 7 h; n = 106,192 cases), long sleep (≥ 9 h; n = 34,184 cases) and normal sleep (7-8 h; n = 305,742 controls)”;
  14. In lines 131-140, 184-189 and 409-414, the sentences should be simplified;
  15. In line 204, “…(FigureA3). We…”;
  16. Figure A2, invisible fonts;
  17. In lines 316 and 317, “(SA)” and “(TH)”, respectively, should be removed;
  18. In line 441, “Emma et al.(51)” should be replaced by “McLachlan et al. (51)”;
  19. Disproportionately enlarged “Discussion”.

English should be double-checked.

Moderate editing of English language required

Author Response

  1. Absolutely unclear through the text, whether participants were all healthy persons or not;

Thank you for your question. The data sources of the ENIGMA consortium are meta-GWAS analyses of normal brain MRI data from 51 cohorts, This study examined the surface area and average thickness of the entire cortex and 34 specialized cortical regions.

  1. Abstract should be simplified by removing of the numerical data and “(SA)”;

In line 23, “TH” should be replaced by “cortical thickness (TH)”;

  1. In line 29, “…parahippocampal brain areas….” and “…the middle temporal zone…”;
  2. In lines 28-30, the statement is incorrect and should be rewritten;
  3. In line 49, “…decline. Measurements of brain cortical surface area (SA) and its TH…”;
  4. In line 50 and everywhere below, references in brackets should be included separately;
  5. In line 60, “…in the carriers (10). “;
  6. In line 67, “…require...” ;
  7. In lines 74 and 76, “IVs” and “SNPs”, respectively, should be open;
  8. In line 80, just “SA”;
  9. In line 93, “exposure” should be clarified;
  10. Figure 1, “SNP”, “GWAS”, “MR-PRESSO” and “n=” should be described in the legend;
  11. In line 103,104, “These 34 regions…” needs to be clarified;
  12. In line 108, “…of TH and SA…” should be removed;
  13. In lines 121-123, “We used the GWAS database of Dashti HS, et al. (17) for short sleep (< 7 h; n = 106,192 cases), long sleep (≥ 9 h; n = 34,184 cases) and normal sleep (7-8 h; n = 305,742 controls)”;
  14. In lines 131-140, 184-189 and 409-414, the sentences should be simplified;
  15. In line 204, “…(FigureA3). We…”;
  16. Figure A2, invisible fonts;
  17. In lines 316 and 317, “(SA)” and “(TH)”, respectively, should be removed;
  18. In line 441, “Emma et al.(51)” should be replaced by “McLachlan et al. (51)”;
  19. Disproportionately enlarged “Discussion”.

English should be double-checked.

Thank you very much for your careful review and constructive suggestions with regard to our manuscript. Meanwhile, I am also very grateful to you for the specific modification strategy.I have made comprehensive and detailed changes in the paper according to your suggestions

Round 2

Reviewer 2 Report

The authors have addressed my all my comments well. Well done!

Author Response

Thank you for your careful guidance!

Reviewer 3 Report

biomedicines-2487044: “Sleep traits causally affect the brain cortical structure: A Mendelian randomization study”.

The authors have made a careful revision and responded to almost all points I raised.

Remarks:

1.     In line 51, “…decline and the measurements of brain cortical surface area (SA) and its TH…”;

2.     In line 52 and everywhere below, the references in brackets need space in front of them;

3.     In lines 142-145, grammatically confusing sentence;

4.     In line 457, “… was associated …”.

Author Response

Thank you for your careful guidance! All changes have been highlighted in the text. Spaces have been added before references in the text.